# Two-Temperature Magnetohydrodynamics Simulations of Propagation of Semi-Relativistic Jets

**Takumi Ohmura [1,\*], Mami Machida [1] , Kenji Nakamura [2] , Yuki Kudoh [3], Yuta Asahina [4] and Ryoji Matsumoto [5]**

1   Graduate School of Science, Kyushu University, 744 Motooka Nishi-ku, Fukuoka 819-0395, Japan; mami@phys.kyushu-u.ac.jp
2   Department of Mechanics, Faculty of Science and Technology, Kyushu Sangyo University, 2-3-1 Matsukadai, Higashi-ku, Fukuoka 813-8503, Japan; nakamura@ip.kyusan-u.ac.jp
3   Graduate School of Science and Engineering, Kagoshima University, Kagoshima 890-0065, Japan; k5751778@kadai.jp
4   Yukawa Institute for Theoretical Physics, Kyoto University Kitashirakawa Oiwakecho, Sakyo-ku, Kyoto 606-8502, Japan; asahina@cfca.jp
5   Department of Physics, Graduate School of Science, Chiba University, 1-33 Yayoi-cho, Inage-ku, Chiba 263-8522, Japan; matsumoto.ryoji@faculty.chiba-u.jp
\*   Correspondence: ohmura@phys.kyushu-u.ac.jp

**Abstract:** In astrophysical jets observed in active galactic nuclei and in microquasars, the energy exchange rate by Coulomb collision is insufficient for thermal equilibrium between ions and electrons. Therefore, it is necessary to consider the difference between the ion temperature and the electron temperature. We present the results of two-temperature magnetohydrodynamics(MHD) simulations to demonstrate the effects of Coulomb coupling. It is assumed that the thermal dissipation heats only ions. We find that the ion and electron temperatures are separated through shocks. Since the ion entropy is increased by energy dissipation at shocks and the Coulomb collisions are inefficient, electron temperature becomes about 10 times lower than the ion temperature in the hotspot ahead of the jet terminal shock. In the cocoon, electron temperature decreases by gas mixing between high temperature cocoon gas and low temperature shocked-ambient gas even when we neglect radiative cooling, but electrons can be heated through collisions with ions. Radiation intensity maps are produced by post processing numerical results. Distributions of the thermal bremsstrahlung radiation computed from electron temperature have bright filament and cavity around the jet terminal shock.

**Keywords:** astrophysical jets; magnetohydrodynamics (MHD); numerical simulation

## 1. Introduction

Astrophysical jets are highly collimated outflows from celestial objects. In this paper, we study the astrophysical jets produced near compact objects such as black holes or neutron stars. Radio observations have revealed that these jets consist of beams, knots, hotspots, and radio-lobes [1]. Multi-wavelength observations at radio, optical, and X-ray frequencies have enabled us to study the emission mechanisms of spatially resolved jets. The radiation from these jets is mainly produced by synchrotron radiation and inverse Compton scattering by non-thermal electrons accelerated in collisionless shocks [2], but thermal emission contributes to emission from the cocoon surrounding the jet beam [3]. The X-ray emission from the cocoon indicates that the electron temperature in this region should exceed $10^7$ K [4].

The propagation of jets in ambient media has been studied theoretically and numerically since the 1970s. Norman et al. [5] conducted two-dimensional hydrodynamic (HD) simulations of the

propagation of light supersonic jets, which exhibited the basic structures of jets such as a bow shock, a terminal shock (a reverse shock), internal shocks, and a cocoon. Recently, jet propagation and its feedback to inter cluster medium has been studied in realistic environments of galaxy cluster by Guo et al. [6,7]. The spacial variation of the propagation velocity of the jet has been reported by analytical methods (e.g., [8,9]) and by numerical simulations (e.g., [10]).

The magnetic field structure of jets has been studied using polarized radiation. Faraday tomography have revealed that jets in 3C273 and SS433/W50 [11,12] have a helical magnetic field. The effects of magnetic fields on the jet dynamics have been examined for both non-relativistic and relativistic jets assuming axisymmetry [13–15]. In three-dimensional calculations, jets develop current-driven kink instability ($m = 1$) [16,17]. In addition, some studies compared numerical results with observations [18–20].

Previous studies, however, did not focus on thermal non-equilibrium between ions and electrons in optically thin plasmas. When the relaxation time of ions and electrons by Coulomb collisions is longer than the dynamical time, the electron temperature can be lower than the ion temperature partly because electrons are cooled by radiation, and partly because shock waves primarily heat ions. Therefore, the electron temperature becomes lower than the ion temperature under such circumstances [21,22].

Such a two-temperature plasma has been studied extensively in the context of optically thin hot accretion flows in galactic black hole candidates (Shapiro, Lightman and Eardley [23]) and in the jet launching region in low luminosity active galactic nucleis (AGNs). Ressler et al. [24] and Sadowski et al. [25] carried out two-temperature magnetohydrodynamics (MHD) simulations in hot accretion flows. They showed that the electron temperature is several tens of times lower than the ion temperature in radiatively inefficient accretion disks. Although the temperature ratio of the ions and the electrons strongly depends on the microphysics in the jets, some studies show that the electron thermal energy is less than or equal to the ion thermal energy [26,27]. These theoretical results support the existence of a two-temperature plasma. Therefore, the evolution of jets should be studied by applying the two-temperature treatment.

In this paper, we present results of magnetohydrodynamics simulations of jet propagation conducted under the two-temperature treatment to study the electron temperature distribution. We examine the influence of Coulomb coupling on the electron temperature distribution, the single temperature results will be compared with the two-temperature results.

This paper is structured as follows. In Section 2, the two-temperature MHD equations and simulation methods are described. Numerical results are presented in Section 3. We discuss the results in Section 4, and give our summary in Section 5.

## 2. Numerical Method

### 2.1. Basic Equations

In our work, we assume pure hydrogen plasma. The non-relativistic two-temperature MHD equations are as follows [28]:

$$\frac{\partial \rho}{\partial t} + \nabla \cdot (\rho \boldsymbol{v}) = 0, \tag{1}$$

$$\frac{\partial (\rho \boldsymbol{v})}{\partial t} + \nabla \cdot \left( \rho \boldsymbol{v}\boldsymbol{v} - \boldsymbol{B}\boldsymbol{B} + p_{\text{gas}} + \frac{\boldsymbol{B}^2}{2} \right) = 0, \tag{2}$$

$$\frac{\partial E}{\partial t} + \nabla \cdot \left[ \left( E + p_{\text{gas}} + \frac{\boldsymbol{B}^2}{2} \right) \boldsymbol{v} - \boldsymbol{B} \left( \boldsymbol{B} \cdot \boldsymbol{v} \right) \right] = 0, \tag{3}$$

$$\frac{\partial \boldsymbol{B}}{\partial t} = \nabla \times (\boldsymbol{v} \times \boldsymbol{B}), \tag{4}$$

$$n T_{\text{e}} \frac{d s_{\text{e}}}{d t} = +\dot{q}^{\text{ie}}, \tag{5}$$

where $\rho$ is the total density of the gas defined by $\rho = \rho_i + \rho_e = n(m_i + m_e) \sim nm_i$, $n_{i,e}$, $\rho_{i,e}$, and $m_{i,e}$ are the number densities, the mass densities, and the mass of charged particles, respectively. Here, the subscripts i and e denote ion and electron, respectively. The temperature, the velocity, and the magnetic fields are denoted by $T$, $v$, and $B$, respectively. The energy transfer rate from ions to electrons through Coulomb coupling is denoted by $\dot{q}^{ie}$. The total gas pressure is given by $p_{gas} = p_i + p_e$. The total energy density $E$ is given by

$$E = \epsilon_i + \epsilon_e + \frac{1}{2}\rho v^2 + \frac{B^2}{2}. \tag{6}$$

Assuming an ideal gas, the internal energy of the ions and electrons are given by

$$\epsilon_i = \frac{p_i}{\gamma_i - 1}, \quad \epsilon_e = \frac{p_e}{\gamma_e - 1}, \tag{7}$$

respectively, where $\gamma$ is the specific heat ratio. We choose a constant specific heat ratio; the ion is non-relativistic ($\gamma_i = 5/3$) and the electron is relativistic ($\gamma_e = 4/3$). The electron entropy per particle can be written in a simple form, $s_e = (\gamma_e - 1)^{-1} \log(p_e n^{-\gamma_e})$. The internal energy of the gas mixture is the sum of the electron internal energy and the ion internal energy. Therefore, the effective gas temperature and specific-heat ratio are given by

$$T_g = \frac{1}{2}(T_i + T_e), \tag{8}$$

$$\gamma_{gas} = 1 + (\gamma_e - 1)(\gamma_i - 1)\frac{1 + T_i/T_e}{(\gamma_i - 1) + (\gamma_e - 1)T_i/T_e}, \tag{9}$$

respectively. The effective specific-heat ratio of the gas mixture is a function of the temperature ratio $T_i/T_e$ and is bounded between $4/3$ and $5/3$.

The energy transfer rate, $\dot{q}^{ie}$, from ions to electrons per unit volume through Coulomb collisions is [29,30]

$$
\begin{aligned}
\dot{q}^{ie} = {} & \frac{3}{2}\frac{m_e}{m_i}n^2\sigma_T c \ln\Lambda(k_B T_i - k_B T_e) \\
& \times \begin{cases} \dfrac{1}{K_2(1/\theta_e)\,K_2(1/\theta_i)}\left[\dfrac{2(\theta_e + \theta_i)^2 + 1}{\theta_i + \theta_e}K_1\left(\dfrac{1}{\theta_m}\right) + 2K_0\left(\dfrac{1}{\theta_m}\right)\right] & (\theta_i > 0.2) \\ \dfrac{\sqrt{2\pi} + \sqrt{\theta_i + \theta_e}}{\theta_i + \theta_e} & (\theta_i < 0.2) \end{cases},
\end{aligned}
\tag{10}
$$

with $\theta_m = \theta_i\theta_e/(\theta_i + \theta_e)$. Here, $\theta_i$ and $\theta_e$ are dimensionless ion and electron temperature defined as Equation (11):

$$\theta_i = \frac{kT_i}{m_i c^2} \quad \text{and} \quad \theta_e = \frac{kT_e}{m_e c^2}. \tag{11}$$

The Thomson scattering cross section is $\sigma_T$ and the Coulomb logarithm is $\ln\Lambda$, which is about 20. Modified Bessel functions of the second kind of the i-th order are denoted by $K_i$.

We assumed axisymmetry and solved Equations (1)–(5) by the conservation form in the $(r, z)$ plane (see Appendix A). The azimuthal components of the vectors are taken into account. We developed a two-temperature MHD code on the basis of CANS+ [31]. CANS+ adopts the HLLD Riemann solver [32], a fifth-order-monotonicity-preserving interpolation scheme [33] and the hyperbolic divergence cleaning method for magnetic fields [34]. We solved the energy equation (Equation (3)) in the conservation form to compute the entropy generation at the shock front. Assuming that all the dissipated energy at the shock front is transferred to ions, we solved the entropy equation for electrons using Equation (5). Here, we followed the method adopted by Kudoh and Hanawa [35] to simulate the two component fluid including thermal plasma and cosmic rays, in which they assume that the cosmic ray fluid is adiabatic at the shock front. We neglected electron dissipative heating at the shock front in this paper.

This is a good approximation for HD shocks because the bulk kinetic energy of ions is one thousand times that of electrons. In contrast, electrons can instantaneously be heated around high Mach number collisionless MHD shocks (e.g., Matsumoto et al. [36]). Indeed, this heating process may affect the electron temperature distribution. In this paper, however, we assume that only the ion thermal energy is increased by shocks to focus on the effect of Coulomb coupling. The Coulomb coupling term on the right-hand side of Equation (5) is computed time explicitly by applying the operator split method.

## 2.2. Numerical Model

A supersonic jet propagating in the $z$-direction is injected at $z = 0$. We assume that jet injection continues during simulation time. The initial jet radius is $r_0$. The jet velocity and Mach number are $v_{jet} = 0.38c$ and $M_{jet} = 6$, respectively. We assumed that the initial ion and electron temperatures are equal ($T_{e,jet} = T_{i,jet} = 1.5 \times 10^{10}$ K). The density ratio of the jet and the ambient medium, $\eta$, is 0.1. The jet and the ambient medium is assumed to be in pressure equilibrium. Thus, the ambient temperature is about $1.5 \times 10^9$ K. We assume that the initial jet beam has the purely toroidal magnetic field $B_\phi = b_{in} \sin^4(2\pi r/r_0)$. The plasma $\beta \equiv p_{gas}/p_{mag} = 2\, p_{gas}/b_{in}^2$ is 10 at $r = 0.5r_0$. We assume an optically thin plasma. The Thomson optical depth of the jets is $\tau = \kappa_e \rho_0 r_0 = 10^{-3}$, where $\kappa_e$ is the electron scattering opacity. We adopted $\kappa_e = 0.38$ cm$^2$g$^{-1}$.

For normalization, we adopted $r_0 = 0.01$ AU, the ambient gas density $\rho_0 = 1.67 \times 19^{-14}$ g/cc, and ambient sound speed $c_{s,0} = 0.02$ c, respectively. The normalization time is $t_0 = r_0/c_{s,0} = 250$ s. The jet parameters are summarized in Table 1. The computation domain is $(r, z) = (40r_0, 160r_0)$ and the number of grids is $(N_r, N_z) = (1024, 2048)$. We adopt uniform grids in both the $r$ and $z$ directions. We apply the reflection boundary condition at the axis of the jet and the zero-gradient boundary at $r = 40r_0$ and $z = 160r_0$. We permit the backflow to escape from the boundary at $z = 0$. The absorbing boundary condition is applied at $z = 0$ and $r > r_0$.

We conducted simulations for four models shown in Table 2. For models TWC and TWOC, Equations (1)–(5) are solved to investigate the effects of Coulomb coupling. Model TWC includes the Coulomb coupling term on the right-hand side of Equation (5) and model TWOC ignores it. We present the results for single-temperature calculations conducted by solving Equations (1)–(4). In the single-temperature models, the gas temperature, the internal energy, and the specific-heat ratio are approximated by the ion values. Model ST1 assumes equal electron and ion temperatures. Model ST10 assumes an electron temperature ten times lower than the ion temperature.

**Table 1.** List of jet parameters.

| Parameter | Value |
|---|---|
| $\tau$ | $10^{-3}$ |
| $\eta$ (=$\rho_{jet}/\rho_{amb}$) | 0.1 |
| $v_{jet}$ | 0.38 c |
| $M_{jet}$ | 6 |
| $T_{jet,i,e}$ | $1.5 \times 10^{10}$ K |
| $T_{amb,i,e}$ | $1.5 \times 10^9$ K |
| $\beta_{jet}$ (=$2p_{jet}/B^2$) | 10 |

**Table 2.** Parameters of numerical models.

| Model Name | Approximation | Coulomb Coupling | $\gamma_i$ | $\gamma_e$ | $T_i/T_e$ | Eqs. |
|---|---|---|---|---|---|---|
| TWC | Two-temperature | Yes | 5/3 | 4/3 | Self consistent | (1)–(5) |
| TWOC | Two-temperature | No | 5/3 | 4/3 | Self consistent | (1)–(5) |
| ST1 | Single-temperature | - | 5/3 | - | 1 | (1)–(4) |
| ST10 | Single-temperature | - | 5/3 | - | 10 * | (1)–(4) |

-: The gas temperature and the specific-heat ratio are approximated by the ion values in One-fluid models.
*: The electron temperature of ST10 is calculated by postprocesing the ion temperature of ST1.

## 3. Numerical Results

### 3.1. Jet Dynamics

Figure 1 shows the typical jet propagation results for model TWC at $t = 70t_0$. The panels show snapshots of the density, absolute velocity, magnetic field strength and plasma $\beta$ from top to bottom. The numbers on the left-hand side of the color bar show the density and the ratio of the azimuthal magnetic field and the initial field, and the numbers on the right-hand side of the color bar show the absolute velocity and plasma $\beta$. The jet dynamics and morphology are similar to those of previous works (e.g., [5]). The Coulomb coupling does not affect the jet dynamics because the total energy does not change by Coulomb collisions. We can identify the bowshock ahead of the jet and the internal shocks in the jet beam. In this paper, we call the region between the bowshock and the contact discontinuity the 'shocked-ambient gas'. A cocoon is formed by the backflow between jet beam and the contact discontinuity separating the jet plasma and the ambient plasma. The back flow has a speed of about 0.1 c. When the jet beam is pinched by the backflow, the deflected flow emanating from the pinched region forms oblique shocks, where the jet speed becomes subsonic [10]. These oblique shocks can be regarded as the secondary terminal shocks formed by the breakup of the jet beam [37]. We can identify three terminal shocks at $z = 80r_0$, $z = 100r_0$, and $z = 115r_0$ for model TWC, and we name the shock at $z = 80r_0$ the 'secondary terminal shock'.

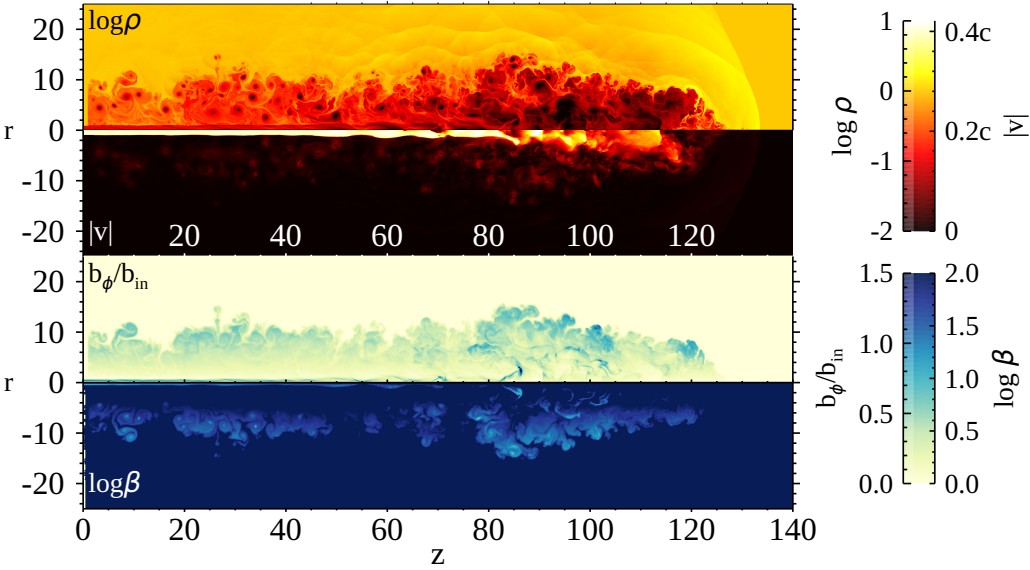

**Figure 1.** Top to bottom: The gas density, the jet absolute velocity, the toroidal magnetic field normalized by the injection field $b_{in}$, and plasma $\beta$ for model TWC at $t = 70t_0$.

The Kelvin–Helmholtz (KH) instability grows between the cocoon and shocked-ambient gas because of velocity shear. The KH instability mixes low-entropy shocked-ambient gas and shocked-heated high-entropy jet gas. The toroidal magnetic fields in the jet beam become stronger ahead of the secondary terminal shock ($z > 80r_0$) as a result of the shock compression. The compressed toroidal fields ahead of the secondary terminal shocks are transported to the cocoon with the backflow. The magnetic fields are stored between the cocoon and the shocked-ambient gas where KH vortices accumulate magnetic flux. The plasma $\beta$ value is around 10 in the cocoon. Therefore, thermal pressure is dominant. Note that poloidal magnetic fields are not generated from the toroidal field because we assumed axisymmetry. When a poloidal field exists, toroidal fields may coil around the surface of the

cocoon. Such a helical magnetic field has been found at the Eastern edge of W50, behind the terminal shock of the SS433 jet [11].

In underdense jets, the propagation speed gradually decreases as the jet propagates. In Figure 2, we plot the time evolution of the bowshock and the terminal shock positions along the $z$-axis for model TWC. The terminal shock is defined as the point closest to $z = 0$ where the beam velocity becomes below 90% of the injection speed. Hence, we identify the secondly terminal shocks after the breakup of the jet beam. We also overplot the analytic results derived by Norman, Winkler and Smarr [38] (dashed) and Zaninetti [9] (solid) in Figure 2. Norman, Winkler and Smarr assume one-dimensional momentum balance between the beam and the ambient gas at the head of jet for the pressure equivalent non-relativistic jet. Results of our numerical simulation agree with the one-dimensional solution (dashed curve) in the early stage $t < 5t_0$. The propagation speed gradually decreases because of the multi-dimensional effects [8,9]. Zaninetti obtained the propagation speed by taking into account the multi-dimensional effects and showed that

$$z \propto t^{3/5}. \tag{12}$$

Our numerical results agree well with Zaninetti's formula, especially at $t < 30t_0$. We find that the propagation velocity of the secondary terminal shock oscillates at $t > 30t_0$.

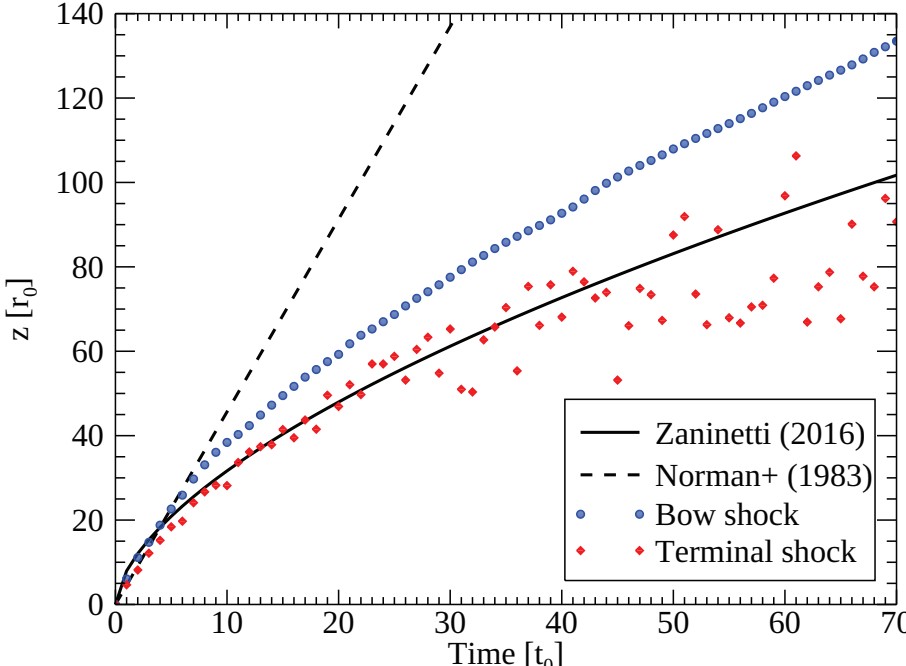

**Figure 2.** Time evolution of the bowshock and the terminal shock positions along the $z$-axis for model TWC. Analytic lines by Norman, Winkler and Smarr (dashed) and Zaninetti (solid) are also plotted.

### 3.2. Electron and Ion Temperature Distributions

Figure 3 shows the distributions of the electron and ion temperatures with a log scale at $t = 70t_0$ for models TWC and TWOC. The electron temperature distributions are significantly different from those of the ion temperature. The temperature in the jet beam increases at internal shocks and jet terminal shocks because energy dissipation occurs at the shocks. Strong ion heating can be seen at the oblique shock at $z = 70r_0$ for model TWC and $z = 85r_0$ for model TWOC. We call the high ion temperature region ahead of the secondary jet terminal shock the hot spot. The temperature increases because of energy dissipation at the secondary terminal shock, and the hot spot is formed between the

secondary terminal shock and the jet head (the contact discontinuity). The propagation speed of the secondary terminal shock is smaller than that of the jet head, and, therefore, the hotspot area increases.

In models TWC and TWOC, the ion temperature in the hot spot is ten times higher than the electron temperature. Electron heating by Coulomb collision is not efficient in the jet beam and the hot spot because electron density is low. In contrast, electron heating via Coulomb collision is important in the cocoon where the electron density is higher than the beam and the hotspot. Figure 1 shows that the gas density in the cocoon increases downstream of the hot spot (smaller $z$). The gas density increases because the KH vortices mix the backflowing plasma and the shocked ambient medium. Since the gas pressure in the cocoon balances with the ambient pressure, ion temperature and electron temperature in the cocoon decrease unless they are heated. Figure 3b shows that, in model TWOC, the electron temperature in the cocoon decreases and becomes comparable to the electron temperature in the shocked ambient medium. On the other hand, in model TWC, electrons can be heated by Coulomb collisions. Therefore, the electrons in model TWC are hotter than those in TWOC in the cocoon. When the radiative cooling is negligible, the Coulomb collision determines the distribution of the electron temperature. The adiabatic expansion does not significantly affect the electron temperature in the cocoon. Since the gas density changes from $0.1\rho_0$ (the initial jet gas density) to $0.01\,\rho_0$ (the cocoon gas density), the temperature reduction rate by adiabatic expansion is $(\rho/\rho_0)^{\gamma_e-1} = 0.1^{1/3} = 0.46$. Since the electron temperature in model TWOC is smaller than this, it indicates that the electron temperature decreases in the cocoon by other mechanisms such as the gas mixing in the cocoon.

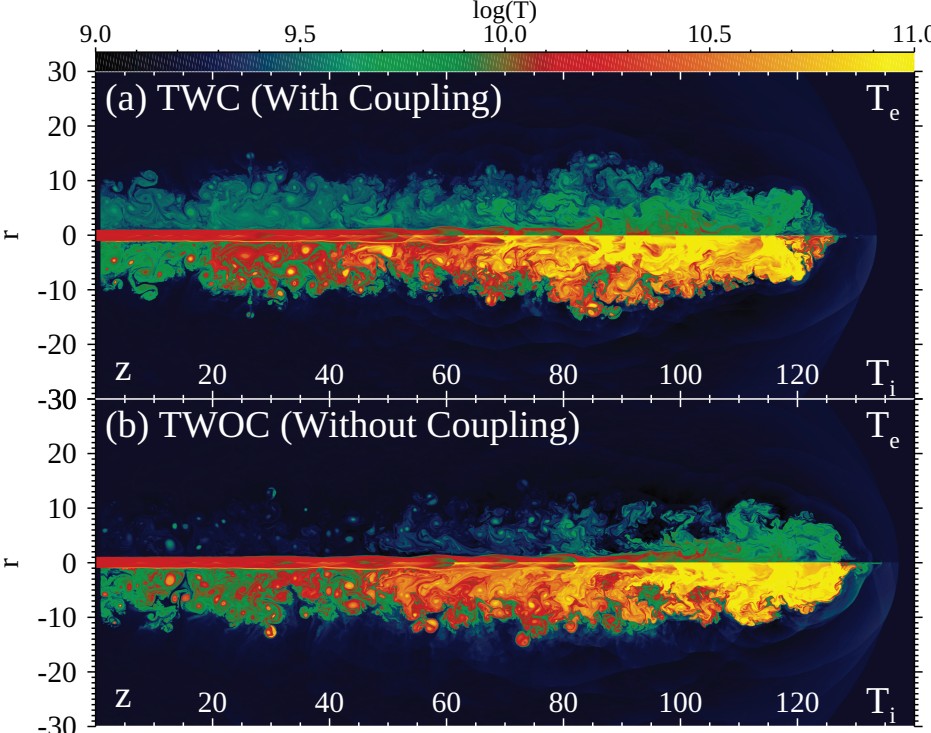

**Figure 3.** (**a**): the distribution of the electron temperature (*top*) and ion temperature (*bottom*) for model TWC at $t = 70t_0$; (**b**): same as (**a**), but for model TWOC.

## 4. Discussion

### 4.1. Comparison with Model ST

In this section, we compare model TWC with model ST. Figure 4 shows the electron temperature distribution for model ST. The upper and lower panels display the results for model ST1 and model ST10, respectively. Figure 5a plots the temperature distribution along the jet beam, and Figure 5b shows

the radial distribution of the temperature averaged in the hotspot ($80r_0 < z < 100r_0$). The curves show the distribution of $T_i$ (red) and $T_e$ (blue) for model TWC, and the electron temperature for model ST1 and model ST10. The temperature distribution for model ST1 is similar to the ion temperature for model TWC. However, the ion temperature for model TWC is 20% higher than the electron temperature for model ST1 in the hotspot because the ions receive all the dissipative energy at the shocks. Therefore, electrons are overheated in model ST1. Conversely, although the electron temperature for model ST10 is a good approximation for electron temperature in the hotspot, the temperature of the cocoon is lower than the electron temperature for model TWC (Figure 3a).

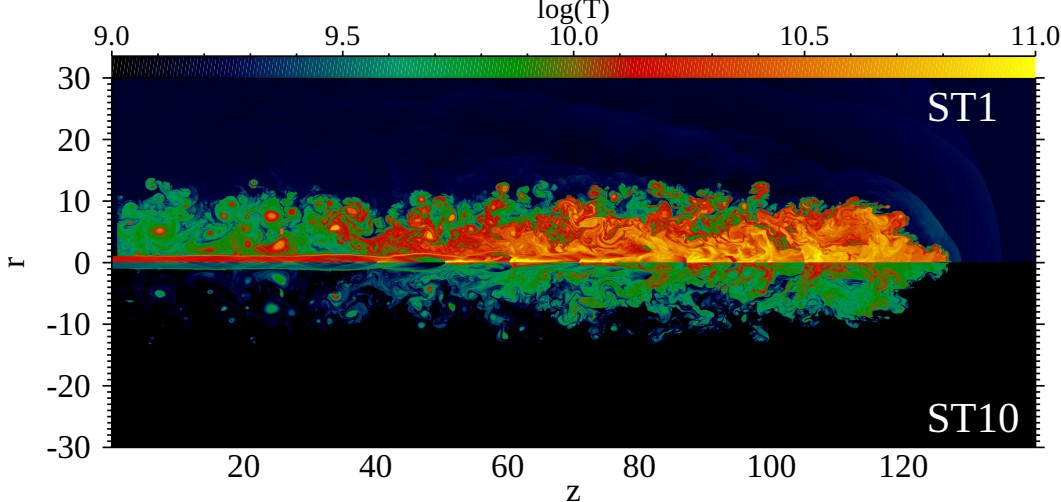

**Figure 4.** The electron temperature distributions with a log scale for model ST1 (*top*) and model ST10 (*bottom*) of model ST at $t = 70t_0$.

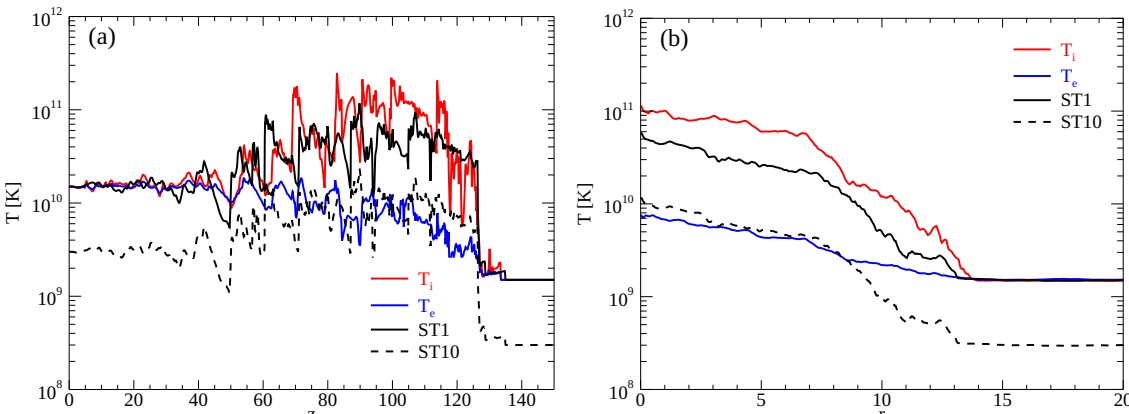

**Figure 5.** Distribution of $T_i$ (red) and $T_e$ (blue) for model TWC, and the electron temperatures for model ST1 (black solid) and model ST10 (black dashed). (**a**): the temperature distribution along the jet axis; (**b**): the radial distribution of the temperature averaged in the hotspot ($80r_0 < z < 100r_0$).

In model TWC, the shocks heat only the ions, and a hotspot is formed in the jet terminal region. The high temperature shocked ion gas is transported to the cocoon, and the electrons with their injection temperature move with ions. In the cocoon, the electron temperature is determined by the balance between heating by Coulomb coupling and cooling by gas mixing. We find that the electron temperature stays around $10^{9.5}$ K in the cocoon by Coulomb coupling. In contrast, when solving single-temperature MHD equations, electron temperature is too high unless we assume electron temperature is much less than ion temperature. In the latter case, however, the electron temperature

in the cocoon becomes much less than that in model TWC and TWOC. Therefore, it is inaccurate to estimate the electron temperature distribution from single-temperature calculations. In other words, the single-temperature approximation is inapplicable to study propagations of high Mach number, low density jet.

*4.2. Bremsstrahlung Images of Jets*

In this section, we calculate radiation intensity maps of the thermal bremsstrahlung radiation. Only the thermal bremsstrahlung is considered. The bremsstrahlung emissivity $\varepsilon_{\text{brmes}}$ is assumed to be proportional to $\rho^2\sqrt{T_e}$. We follow the analysis of Scheck et al. [39]. The radiation intensity map is obtained by three steps. First, we calculate the bremsstrahlung emissivity. Second, we convert the 2D axisymmetric cylindrical coordinates to 3D Cartesian coordinates. Finally, we sum the emissivity along the line of sight direction (the viewing angle is $90°$ from the jet axis):

$$E_{\text{brems}} = \int \epsilon_{\text{brems}}(x, y, z) dy. \tag{13}$$

Figure 6a shows the radiation intensity map at $t = 40t_0$ for model TWC. The radiation from the ambient gas and the shocked-ambient gas is stronger than that from the jet because bremsstrahlung emissivity is proportional to the square of the density. The radiation intensity map is similar to the radio image of the ring around Cyg X-1 [40]. The radio ring around Cyg X-1 is considered to be due to the bremsstrahlung radiation. However, the ambient density and pressure in model TWC is too large except the region close to the central compact object. The ambient temperature of the radio emitting region in Cyg X-1 is around $10^4$ K, which is much lower than $T \sim 10^{9.5}$ K in model TWC. We expect that the temperature after propagation longer than the end of simulation will be the same as the observed temperature.

Therefore, we exclude the bremsstrahlung emission from the ambient matter following the procedures adopted by precedents (e.g., [10,39]). Figure 6b shows the radiation intensity maps computed by setting $\epsilon_{\text{brems}} = 0$ when $T_e < T_{e,\text{amb}}$ at $t = 40t_0$ (*top*) and $t = 70t_0$ (*bottom*). The black contours in both panels show $z$-velocities of 0.38c, 0.34c, and 0.32c. At $t = 40t_0$, the size of the hot spot between the jet secondary terminal shock and the contact discontinuity separating the ambient plasma and jet plasma is small. The jet head which corresponds to the hotspot in AGN jet observations is the brightest. The cavity is formed behind the strong terminal shock. Since the mean density of the cocoon increases as $z$ decreases, the jet is brighter near the injection point of the jet. The electron temperature remains higher than ambient temperature via Coulomb collision. Since the mean density in the cocoon becomes higher, the intensity increases with decreasing $z$. At $t = 70t_0$, the jet is brightest at the jet head. The hotspot size grows with time, and intensively radiating filament appears in the hotspot. Moreover, the radiation for model TWC is brighter than that at $t = 40t_0$ around the root of jet.

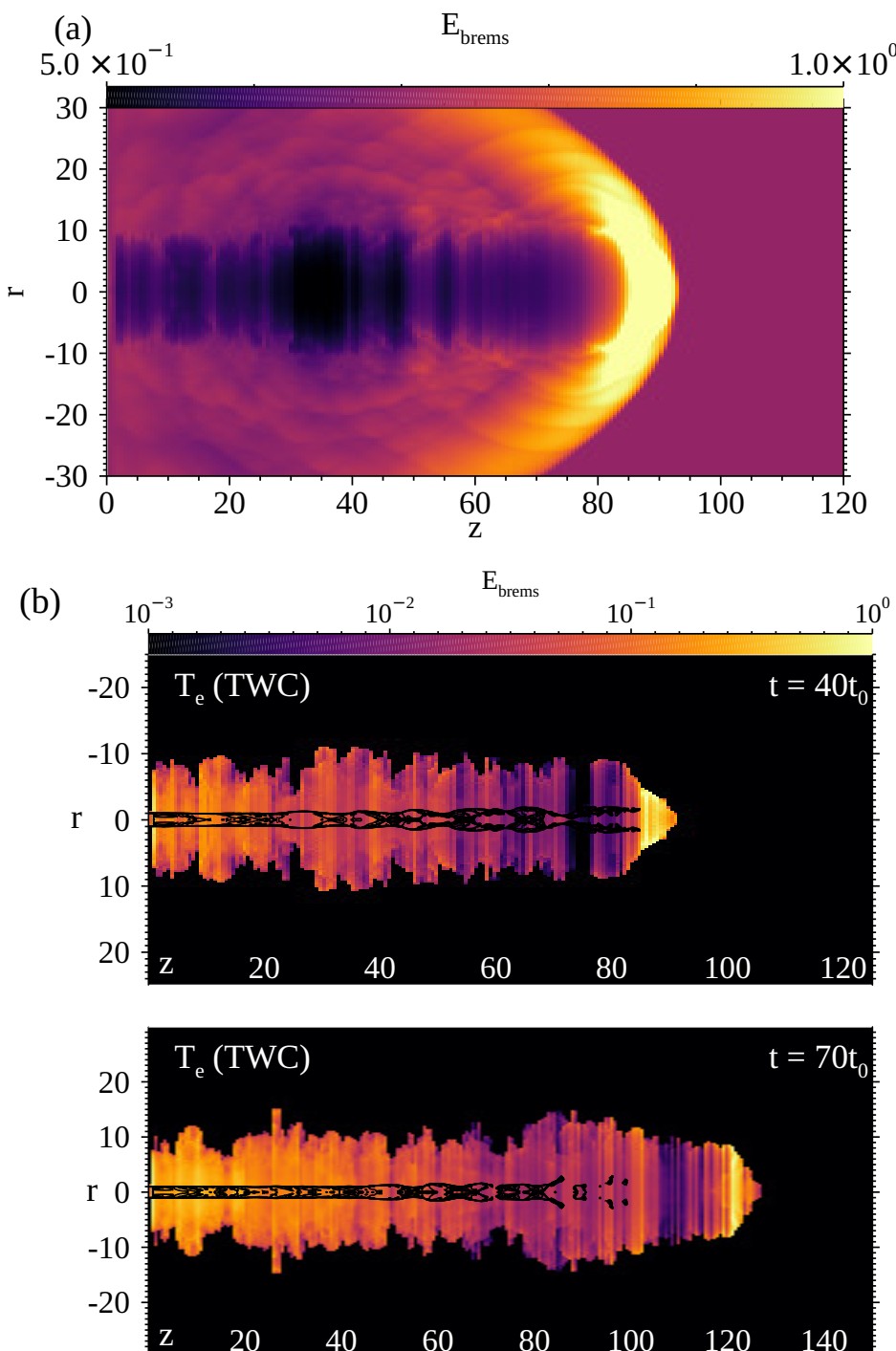

**Figure 6.** (**a**) the radiation intensity map at $t = 40t_0$ for model TWC; (**b**) the radiation intensity maps of the jet and cocoon computed by setting $\epsilon_{brems} = 0$ when $T_e < T_{e,amb}$ at $t = 40t_0$ (*top*) and $t = 70t_0$ (*bottom*). The black contours show the contour of *z*-velocities of 0.38 c, 0.34 c, and 0.32 c.

### 4.3. Electron Heating through the Various Physical Mechanisms

In this section, let us discuss the dependence of numerical results on heating and cooling processes. In the simulation presented in this paper, we considered shock heating of ions by solving the conservation form of energy equations combined with the equation for electron entropy taking into account only the Coulomb collision. We adopted this approach to illuminate the effects of the

Coulomb coupling. In astrophysical plasma, magnetic reconnection and turbulent cascade of energy can heat not only ions but electrons. The fraction of the various dissipative electron heating, $f_e$, is determined by microphysical processes. Preceding works for two-temperature MHD simulations of hot accretion flows [26,41] adopt a sub-grid prescriptions, which are based on the results of the gyrokinetics simulations (H10) [42] and particle-in-cell (PIC) simulations of fast magnetic reconnection (R17) [43], to calculate $f_e$. Kawazura, Barnes and Schekochihin [44] carried out numerical simulation using a hybrid fluid-gyrokinetic model, and updated the results of H10 (K18).

We estimate the electron heating rate using model K18 and R17. First, the K18 model shows a strong dependence of the ratio of ion pressure to the magnetic pressure $\beta_i$. In our simulation, since $\beta_i \sim 10$ everywhere in jets, we expect that the heating ratio $Q_i/Q_e$ is greater than 10 using Equation (2) in K18. Therefore, more than 90% dissipated energy will heat ions.

Next, we discuss for the magnetic reconnection model R17. The heating fraction $f_e$ is sensitive to the magnetization parameter, $\sigma_w$ defined as the ratio of fluid enthalpy and magnetic energy. Since ion gas is non-relativistic in the regime of interest for our simulations ($T_i \sim 10^{10}$–$10^{11}$ K), non-relativistic reconnection heating rate is $f_e = 0.14$ independent of $\beta_i$. Therefore, most of the dissipation energy goes to the ions in both models. In summary, our main result that electrons are mainly heated by Coulomb coupling do not change.

We further discuss instantaneous electron heating at shocks. Since efficiency of electron heating strongly depends on Mach number, upstream plasma $\beta$, and the angle between magnetic field and the shock normal vector, the post-shock electron temperature is hard to be determined uniquely. Guo et al. [45,46] obtained the post-shock electron temperature in perpendicular low Mach number shocks ($4 < \beta_i < 32$ and $2 < M_s < 5$ where $M_s$ is sonic Mach number). These results indicate that the ratio of electron and ion temperatures, $T_e/T_i$, decreases in proportion to the sonic Mach number downstream of the shock, and the temperature ratio becomes much smaller than 0.2 when $M_s > 5$. Therefore, we anticipate that electron heating is insufficient at shocks because the shocks are $\beta \sim 10$ and $M_s \sim 6$ in our simulations.

## 5. Conclusions

We have carried out two-dimensional, two-temperature MHD jet propagation simulations to clarify the effects of different ion and electron temperatures. Our simulations are based on the kinetic-energy dominant jet model (e.g., Norman et al. [5]).

Our main result is that the electron temperature is 10 times lower than the ion temperature in the hotspot and the cocoon because the ion entropy is increased by energy dissipation at shocks and the Coulomb collisions are inefficient. In the downstream region of the cocoon, the mean gas density becomes higher because KH instability mixes the shocked ambient gas and the gas in the cocoon. Since the Coulomb collision rate is enhanced in the cocoon, the electrons can stay hot ($T_e \sim 10^{9.5}$ K) by energy transfer from ions to electrons via Coulomb collisions. Because of the lack of radiative cooling, the jet propagation is similar for both the one-temperature and two-temperature models We compare the propagation speed of our simulation results with the analytic results. Our numerical results agree well with the analysis by Zaninetti [9]. The magnetic fields in the jets are amplified by the compression at the secondary terminal shock, and the magnetic fields are stored between the cocoon and the shocked-ambient gas. These coiling magnetic fields have been observed using polarization analysis at radio wavelengths in W50/SS433 eastern ear [11].

The intensity maps of the bremsstrahlung radiation demonstrate the importance of the two-temperature treatment. The jet is brightest at head, and bremsstrahlung radiation is dominant in the downstream region of the cocoon because the electrons can stay in high temperatures by Coulomb collisions. A low density region appears behind the jet terminal shock.A bright region appears in the jet head between the jet terminal shock and the contact discontinuity separating the jet and the ambient medium. Since the distance between the terminal shock and the contact discontinuity increases with time, the size of the hot spot increases. Therefore, radiation cavity appears around the jet terminal

shock at early simulation times. The hotspot size grows with time, and intensively radiating filament appears around the hotspot.

Numerical results depend on heating and cooling processes, especially on the electron heating, which depends on microphysics. We considered three electron heating models which are the turbulent damping model (K18) based on the gyrokinetics simulations, the fast magnetic reconnection model (R17), and the shock heating model. All three models indicate that ions are heated more efficiently than electrons unless the low-beta jets are injected.

Radiative cooling, bremsstrahlung radiation, synchrotron radiation, and inverse Compton scattering are ignored in our work, but these effects are important for jet dynamics and comparisons with observations. We should consider the following factors: the effect of the background environment, 3D effects, and the effect of dissipative heating at scales smaller than the MHD grid-scale. Our ultimate goal is to obtain simulation results that can be directly compared with observations of radio and X-ray jets.

**Author Contributions:** T.O. and Y.K.; Methodology, T.O.; Software, T.O.; Writing—Original Draft Preparation, M.M., K.N., Y.A. and R.M.; Writing—Review and Editing, M.M.and R.M.; Funding Acquisition, M.M.; Supervision.

**Funding:** This work was supported by JSPS KAKENHI Grant Number 16H03954.

**Acknowledgments:** We acknowledge useful discussions and suggestions regarding our numerical methods from Y. Matsumoto, and we thank the anonymous referees for useful comments and suggestions that have led to a substantially improved manuscript. Our numerical computations were carried out on the Cray XC50 and XC30 at the Center for Computational Astrophysics in the National Astronomical Observatory of Japan. This research is partially supported by the Initiative on the Promotion of Supercomputing for Young or Women Researchers of the Information Technology Center at the University of Tokyo. We thank Edanz Group (www.edanzediting.com/ac) for editing a draft of this manuscript.

**Conflicts of Interest:** The authors declare no conflict of interest.

## Appendix A. Conservation Form

The two-temperature MHD Equations (1)–(5) can be arranged in conservation form as follows:

$$\frac{\partial \boldsymbol{U}}{\partial t} + \nabla \cdot \boldsymbol{F} = \boldsymbol{S}, \tag{A1}$$

$$\boldsymbol{U} = \begin{pmatrix} \rho \\ \rho v \\ \boldsymbol{B} \\ e \\ \rho s_e \end{pmatrix}, \boldsymbol{F} = \begin{pmatrix} \rho v \\ \rho v v + p_T \boldsymbol{I} - \boldsymbol{B}\boldsymbol{B} \\ v\boldsymbol{B} - \boldsymbol{B}v \\ (e + p_T)v - \boldsymbol{B}(v \cdot \boldsymbol{B}) \\ \rho s_e v \end{pmatrix}, \boldsymbol{S} = \begin{pmatrix} 0 \\ 0 \\ 0 \\ 0 \\ m_i \dot{q}_{ie}/T_e \end{pmatrix}, \tag{A2}$$

where $\boldsymbol{U}$ and $\boldsymbol{F}$ denote the state and flux vectors, $\boldsymbol{I}$ is a unit matrix, $\boldsymbol{S}$ is source term, and total energy, $e$, is given by

$$e = \epsilon_i + \epsilon_e + \frac{1}{2}\rho v^2 + \frac{1}{2}B^2, \tag{A3}$$

$p_T = p_i + p_e + p_{mag}$ is total pressure and $s_e$ is the electron gas's entropy. The electron gas entropy per particle is given by

$$s_e = (\gamma_e - 1)^{-1} \log\left(p_e n^{-\gamma_e}\right). \tag{A4}$$

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
