# Peer review of "Two-Temperature Magnetohydrodynamics Simulations of Propagation of Semi-Relativistic Jets"

_galaxies, doi:10.3390/galaxies7010014_

Round 1
Reviewer 1 Report
This paper presents axially symmetric simulations of magnetised relativistic jets, focussing on the effects caused by the different temperature between electrons and ions in the fluid. It addresses an interesting topic, nevertheless I find that this paper needs to undergo some revision before being reconsidered for publication. There are two major issues that needs to be addressed which I discuss below. Moreover, several statements are either made without any citation to support them. Furthermore, some claims not obvious and probably inconsistent with the results shown in the figures. I discuss the major issues of the paper in section A of this review and further issues in section B.
A. Major issues:
I have a hard to understand which equations are integrated for each one of the models presented. The authors need to state explicitly which equations are solved for each problem. I understand that equations 1-4 are integrated for all models. After that, it is not clear which one among equations describing energy transfer between electrons and ions are taken into account in each one of the models (TWC, TWOC, ST1, ST10).
The presentation of the results in its current form is descriptive and lacks physical explanation and insight. Namely, while the authors identify the differences between the results of different simulations they do not look into which processes are operating and what causes these difference. In order to do that, they will need to further quantify their results estimating what fraction of energy is exchanged between different species through the various physical mechanism.
B. Further issues:
The authors state in the abstract that “The origins of the radiation from astrophysical jets are considered by synchrotron, bremsstrahlung and inverse Compton processes” however in their paper they only consider radiation through bremsstrahlung. This statement needs to be revised.
The authors state in the abstract that: “The maps of one-temperature results show uniform high intensity radiation. In contrast, bremsstrahlung radiation which is estimated by electron temperature is bright on the root of jets.” I couldn’t find how this statement follows by their calculations, namely the jet is bright at hotspot for the TWC model (Figure 5).
A citation is needed to support the statement: “Radiation from jets is mainly by synchrotron radiation and inverse Compton scattering by non-thermal electrons accelerated in collisionless shocks but thermal emission contributes to the emission from the sheath between the jet and the ambient medium.”
A citation is needed to support the statement: “When the relaxation time of ions and electrons by the Coulomb collision is longer than the dynamical time, the electron temperature can be lower than ions because electrons subject to cooling by radiation. On the other hand, a shock primarily heats ions. Therefore, the electron temperature is lower than the is lower than the ion temperature in such circumstances”.
They need to provide a citation for the equations or relativistic MHD.
They also need to explain whether the quantities appearing in their equations and discussion are in the lab or the co-moving fluid frame.
The author state that: “We neglected electron dissipative heating at shock front in this paper”, is this choice justified? How does it affect their results?
The Mach number is set to 6, and the velocity to 0.4c, while the sound speed c_s,0=0.02. These numbers are not consistent.
What is the resolution of the simulations? Did the authors performed any convergence test?
The authors need to explain what b_in (Is the initial maximum value of the magnetic field? Is it the initial local value of the magnetic field?)
What is t_0 measured in light-crossing times of their domain?
There is a problem with numbering of figures. Probably, at the beginning of section 3.2 the authors discuss figure 2 not figure 3.
The authors state “These result, however, should be viewed with a caution as the electron energy did not increase by shock heating.” Was it expected to increase electron energy due to shock? Is this an indication that a physical processes is not modelled properly?
In line 138 do the authors mean Figure 3?
The authors state “This results indicate that electron temperature distribution for model TWC can not directly estimate to assume the ion-to-electron temperature ratio.” Do they assume that model ST10 is more trustworthy than TWC? Where they expecting to recover the same results?
The authors state: “In order to avoid the numerical difficulty of the calculation for the Coulomb coupling term, we assume that the ambient gas relativity higher temperature than ionized plasma.” I don’t understand why then need to make this assumption, in all the results the ambient gas is much cooler than the jet thus its emissivity is going to be much lower than that of the jet.
The author state that “The magnetic fields in jets are amplified by the terminal shock and propagate to the hotspot.” I can’t see from the figures how the magnetic field propagates to the hotspot, actually it is rather high at the axis of the jet, and in several places where the jet has created turbulent plumes.
Author Response
We thank the reviewer for your carefully reading to our manuscript and giving us various useful comments.
We used English editing service to improve English.
Revised part is shown by blue characters in the manuscript.
To respond to the comments from other reviewers, we have added new figures, a new table, and a new discussion section.
Moreover, we have revised the title of this paper to clarify the research theme.
Best regards,
Takumi Ohmura
On the behalf of the authors

Reviewer 2 Report
The paper is interesting and deserves publication.
The following items point towards the readability
of the paper for the occasional reader.
1) English : the paper should be checked by a native speaker
as an example after formula (5) we have densies
rather than densities
2) In the introduction some references to the problem
of the velocity's decrease should be done.
An example is given by the conservation of the
flux for momentum and energy , see Zaninett2015 and Zaninetti2016
in bibliography below
3) The word classic is missing when the basic equations
(1-5) are presented.
For a relativistic treatment of the MHD equations
see equation (2) in Ferrari1980 in bibliography below
4) After formula (7) the adopted values for
gammai and gammae should be justified and references
inserted
5) The equations are classic but the velocity of the
simulation for the jet is relativistic 0.4 *c, see line 71
and Table 1
I would suggest to decrease the velocity in order to be
consistent
6) The behavior of the velocity as function of the distance
is nor clearly outlined , it is constant .. it decreases ?
7) More details should be inserted on the chosen geometry
of the magnetic field
References
@ARTICLE{Ferrari1980,
author = {{Ferrari}, A. and {Trussoni}, E. and {Zaninetti}, L.},
title = "{Magnetohydrodynamic Kelvin-Helmholtz instabilities in
astrophysics. I - Relativistic flows - plane boundary layer in vortex sheet
approximation}",
journal = {\mnras},
keywords = {Astrophysics, Boundary Layer Plasmas, Kelvin-Helmholtz
Instability, Magnetohydrodynamic Stability, Relativistic Plasmas, Vortex
Sheets, Extragalactic Radio Sources, Plasma Jets, Plasma Oscillations},
year = 1980,
month = nov,
volume = 193,
pages = {469-486},
doi = {10.1093/mnras/193.3.469},
adsurl = {http://adsabs.harvard.edu/abs/1980MNRAS.193..469F},
adsnote = {Provided by the SAO/NASA Astrophysics Data System}
}
@ARTICLE{Zaninetti2015,
author = {{Zaninetti},L.},
title = "{Classical and relativistic
conservation of momentum flux
in radio-galaxies
}",
journal = {Applied Physics Research},
year = 2015,
volume = 7,
pages = {43-62}
}
@ARTICLE{Zaninetti2016,
author = {{Zaninetti}, L.},
TITLE = {Classical and Relativistic Flux of
Energy Conservation in Astrophysical Jets},
JOURNAL = {Journal of High Energy Physics, Gravitation and Cosmology},
VOLUME = {1},
YEAR = {2016},
PAGES = {41-56},
}
Author Response

(The authors gave the same response as above.)

Reviewer 3 Report
This paper studies the evolution of astrophysical jets in a uniform ambient medium with 2D MHD simulations. The unique and novel part of this work is to treat ions and electrons separately with different temperatures, while most previous studies treat electrons and ions at the same temperature. As real jets contain very low density plasma where electrons may indeed be not in thermodynamic equilibrium with ions, the topic of the current work should be of great interest to the readers. I think that the paper is of scientific merit to the related astrophysics society, but I do have a few questions and suggestions that the authors should consider to address before recommending it for publication.
1. The main weakness of the paper is English grammar. While the paper should be mostly understandable to the readers, there are lots of errors (mostly minor though) in English grammar. I suggest the authors to read through the manuscript carefully and fix these minor issues (and revise/polish the manuscript). Alternatively, it may be very helpful to ask help from a native English speaker.
2. Page 1, line 10. What does it mean exactly by claiming "the electron can be kept around rest mass energy ..."? I understand that the electrons are heated by Coulomb coupling, but still don't understand what this statement means exactly. By including both rest mass energy and thermal energy, I would expect that the total energy of electrons should always be more than rest mass energy.
3. Page 1, line 17. "block holes" should be replaced by "black holes".
4. Page 2, first paragraph. It may be useful to include a few additional references here while describing previous studies of jet simulations. e.g., jet evolution has been previously studied with hydrodynamic simulations in realistic environments of galaxy clusters in
http://adsabs.harvard.edu/abs/2016ApJ...826...17G
and
http://adsabs.harvard.edu/abs/2018MNRAS.473.1332G
and jet evolution has also been invoked to explain the Fermi bubbles observed in the Milky way in
http://adsabs.harvard.edu/abs/2012ApJ...756..181G
5. Page 2, equation (5). There is the gas density "rho" on the left-hand side of the equation. Should it be "rho" or "rho_e" (i.e. electron mass density)? Furthermore, it would be very helpful to write down explicitly the expression for electron entropy "s_e" in terms of number density, electron temperature, etc.
6. Page 3, line 64. The authors assume that "all the dissipated energy at the shock front goes to ions". I wonder how this is implemented technically in the code. If possible, it would be useful to explain it in a bit more detail.
7. Page 4, line 83. What is the absorbing boundary condition? Why not using the reflective boundary condition at z=0 and r>r_{0}?
8. Page 4, section 2.2. Is the jet always active during the simulations? Or the jet is turned off after a duration during the simulations
9. Page 4, line 90. What is "t_0" here? The value of "t_0" is not given in the paper.
10. Page 5, Figure 1. In the caption, the bottom panel is not described. The caption only mentions gas density, velocity, and field strength (the top three panels).
11. Page 5, line 114. Here "Figure 3" should be replaced by "Figure 2". Am I correct?
12. Page 5, line 119. There must be a typo in this line, and I don't understand what it means. Is there a missing word "between" between "the distance" and " the jet head"?
13. Page 6, line 138. Here "Figure 3b" should be replaced by "Figure 3". Am I correct?
14. Page 6, line 150. "directly estimate to assume the" may be replaced by "be directly estimated by assuming an".
15. Page 6, the paragraph below line 151. The authors assume that the bremsstrahlung emissivity to be proportional to rho T_e^{0.5}. I thought it should be proportional to rho^{2] T_e^{0.5}. Is my understanding correct?
16. Page 6, the paragraph below line 151. I don't understand the sentence "we assume that the ambient gas relativity higher temperature than ionized plasma". I suggest to rephrase it to be better understood.
17. The bremsstrahlung emissivity (or intensity) is sometimes denoted as "brems", while sometimes as "brmes" (e.g. in the title of Figure 5 and the second line below line 151). This is clearly a typo and can be fixed.
18. What is the time of the Figure 4? Is it also "t=70 t_0"? It should be explicitly stated in the caption of this figure.
Author Response

(The authors gave the same response as above.)

Round 2
Reviewer 1 Report
The authors have considered all my comments and they have modified their manuscript appropriately. I suggest the acceptance of their paper for publication.
Reviewer 2 Report
The authors have answered to the points of the first report.